**Data Availability Statement:** All relevant data are within the manuscript.

**Funding:** This work was supported by JSPS KAKENHI Grant Nos. JP19K18807 (principal

# Upregulation of a nuclear factor-kappa B-interacting immune gene network in mice cochleae with age-related hearing loss

**Kensuke Uraguchi, Yukihide Maeda◉\*, Junko Takahara, Ryotaro Omichi, Shohei Fujimoto, Shin Kariya, Kazunori Nishizaki, Mizuo Ando**

Department of Otolaryngology- Head and Neck Surgery, Okayama University Graduate School of Medicine, Dentistry and Pharmaceutical Sciences, Okayama, Japan

\* yamayuki@cc.okayama-u.ac.jp

## Abstract

Epidemiological data suggest that inflammation and innate immunity play significant roles in the pathogenesis of age-related hearing loss (ARHL) in humans. In this mouse study, real-time RT-PCR array targeting 84 immune-related genes revealed that the expressions of 40 genes (47.6%) were differentially regulated with greater than a twofold change in 12-month-old cochleae with ARHL relative to young control mice, 33 (39.3%) of which were upregulated. These differentially regulated genes (DEGs) were involved in functional pathways for cytokine–cytokine receptor interaction, chemokine signaling, TNF signaling, and Toll-like receptor signaling. An NF-κB subunit, *Nfkb1*, was upregulated in aged cochleae, and bioinformatic analyses predicted that NF-κB would interact with the genomic regulatory regions of eight upregulated DEGs, including *Tnf* and *Ptgs2*. In aging cochleae, major proinflammatory molecules, *IL1B* and *IL18rap*, were upregulated by 6 months of age and thereafter. Remarkable upregulations of seven immune-related genes (*Casp1*, *IL18r1*, *IL1B*, *Card9*, *Clec4e*, *Ifit1*, and *Tlr9*) occurred at an advanced stage (between 9 and 12 months of age) of ARHL. Immunohistochemistry analysis of cochlear sections from the 12-month-old mice indicated that IL-18r1 and IL-1B were localized to the spiral ligament, spiral limbus, and organ of Corti. The two NF-κB-interacting inflammatory molecules, TNFα and PTGS2, immunolocalized ubiquitously in cochlear structures, including the lateral wall (the stria vascularis and spiral ligament), in the histological sections of aged cochleae. IBA1-positive macrophages were observed in the stria vascularis and spiral ligament in aged mice. Therefore, inflammatory and immune reactions are modulated in aged cochlear tissues with ARHL.

## Introduction

Age-related hearing loss (ARHL; namely presbycusis) is a major medical and social issue in developed countries with rapidly aging populations. In the United States, approximately one third of the total population aged 65–74 years experiences ARHL, and nearly half of the

investigator: S.F.) and JP20K09732 (S.K.). The Japan Society for the Promotion of Science (www.jsps.go.jp/) had no role in study design, data collection and analysis, decision to publish, or preparation of the manuscript.

**Competing interests:** The authors have declared that no competing interests exist.

population aged >75 years has hearing difficulties [1]. In Japan, which is the most rapidly aging country in the world, persons aged ≥65 years represented 28.1% of the total population in 2018, and this percentage is anticipated to reach approximately 40% by 2065 [2]. ARHL significantly affects the health of older adults, leading to difficulty in communication, mental disabilities such as depression and dementia, low quality of life, and decreased social activity [3]. The progression of ARHL is thought to involve multiple molecular mechanisms in the cochlea; therefore, it is important to elucidate the pathologic mechanisms underlying ARHL in the cochlea so that preventive and therapeutic treatments for ARHL can be developed.

Schuknecht divided the classical human histopathological findings of ARHL in the cochleae into four categories—sensory presbycusis, neural presbycusis, metabolic presbycusis, and conductive cochlear loss—according to the site of abnormalities in the microscopic cellular structures of the cochleae, including the hair cells, stria vascularis, and spiral neurons [4]. Experimental studies in the cochleae of mice with ARHL found that the cumulative effect of oxidative stress damages mitochondrial DNA, and in turn, mutations/deletions in mitochondrial DNA lead to a decline in mitochondrial function and apoptosis in cochlear cells [5, 6].

Epidemiological data suggest that inflammation and innate immunity play significant roles in the pathogenesis of ARHL in humans. Epidemiological studies have reported that elevations in serum C-reactive protein levels, neutrophil counts, and inflammatory cytokine interleukin (IL)-6 levels are associated with a higher risk of ARHL and worse hearing levels in older adults [7, 8]. However, data from animal experiments mechanistically demonstrating the involvement of inflammation and innate immunity in the pathology of ARHL in the cochleae are scarce [9, 10].

As an animal model of ARHL, inbred C57BL/6J mice exhibit the ARHL phenotype as early as 6 months of age [11]. Technically, gene expression studies by means of DNA microarray and next-generation sequencing (RNA-seq) allow genome-wide analyses of approximately 22,000 mice genes; however, real-time RT-PCR outperforms these technologies by enabling more accurate quantification of specific gene expressions encoding proteins with known functions. Therefore, in the first step of this study, the expression levels of 84 inflammatory and immune-related genes were analyzed by a real-time RT-PCR array in the cochlea of 12-month-old and 6–7-week-old C57BL/6J mice, and as many as 33 immune-related genes were found to be upregulated in cochleae of the older mice. In the second step, at which time points of the aging process (3-, 6-, 9-, and 12-month-old C57BL/6J mice) such upregulations of immune-related genes were observed was investigated. In the third step, immunohistochemical experiments were performed to clarify the histological localization of such immune-related gene/protein expressions in mice cochleae. These data help provide a more precise understanding of how the immune process occurs in the cochleae of aging mice with ARHL.

## Materials and methods

### Dissection of mice cochlear tissues and RNA extraction

All animal experiments were performed in compliance with the ethical standards approved by Okayama University's Committee on the Use and Care of Animals (protocol Nos.: OKU-2018847, 2019396, 2019397, 2019398, and 2020549; principal investigator: Y.M.) and adhered to national and international standards of animal care.

For the extraction of cochlear RNA samples to perform PCR-based gene expression experiments, 6–7-week-old [young control mice], 12–13-week-old [3 months], 25–26-week-old [6 months], 38–39-week-old [9 months], and 52–54-week-old [12 months]) male C57BL/6J mice were obtained from Charles River Laboratories (Yokohama, Japan). For the dissection of cochlear samples from mice at 6–7 weeks ($n = 6$) and 3 ($n = 6$), 6 ($n = 6$), 9 ($n = 6$), and 12

months ($n = 6$), deeply anesthetized mice (intraperitoneal ketamine [80 mg/kg] and xylazine [8 mg/kg]) were euthanized via cervical dislocation. The cochlear tissues were promptly dissected into collection tubes containing RNA later reagent (Qiagen, Hilden, Germany). The samples were then incubated in RNA later (Qiagen) at 4˚C for 24 h and stored frozen until RNA purification. The tissues were homogenized, and total RNA was purified using a miR-Neasy mini column (Qiagen). The quantity and quality of the RNA samples were assessed using a spectrophotometer (NanoDrop™ One; Thermo Fisher Scientific, Waltham, MA, USA) and the Agilent 2100 Bioanalyzer system (Agilent Technologies, Santa Clara, CA, USA). One cochlear tissue was collected from a mouse, and >0.6 μg of total RNA with an RNA integrity number >8.0 was purified per one cochlear tissue.

## Hearing levels in mice with ARHL

It was verified that 6- and 12-month-old male C57BL/6J mice exhibited significant ARHL based on click-auditory brainstem response (c-ABR) thresholds compared with younger 6–7-week-old male C57BL/6J mice. Hearing levels were assessed using the c-ABR as previously described [12]. Under anesthesia with intraperitoneal ketamine (80 mg/kg) and xylazine (8 mg/kg), click sounds were delivered to the ear in 5-dB steps from 90 dB sound pressure level (SPL) to 0 dB. c-ABR was recorded by needle electrodes inserted into the vertex and postauricular area as an averaged record of 1000 responses for each SPL. The c-ABR threshold was defined as the minimum SPL at which the c-ABR was clearly recognized. c-ABR thresholds in the young control mice ($n = 7$), 6-month-old mice ($n = 10$), and 12-month-old mice ($n = 8$) were compared using the Kruskal–Wallis and Mann–Whitney $U$ tests.

## Real-time RT-PCR array (RT$^2$ Profiler™)

Between the cochleae of 12-month-old and 6–7-week-old mice, the differences in expression levels of 84 key genes actively involved in inflammatory and immune functions were analyzed using the RT$^2$ Profiler™ PCR array (Qiagen). The RT$^2$ Profiler™ PCR array is a 96-well plate spotted with specific primers for 84 targeted genes to each well. The targeted genes were those encoding cytokines (including chemokines and interleukins), their receptors and signaling molecules, and genes involved in acute, chronic, and intracellular inflammatory responses. The entire list of the 84 genes analyzed in the array is provided as S1 Table.

A cDNA library was synthesized by reverse-transcription of the RNA samples (500 ng) from the 12-month-old and 6–7-week-old cochleae using the RT$^2$ First Strand Kit (Qiagen). With the cDNA libraries used as templates for the PCR reactions, the expression levels of 84 key genes related to inflammatory and immune functions were profiled by the mouse inflammation and autoimmunity RT$^2$ Profiler™ PCR array (PAMM-077Z; Qiagen) using the Light-Cycler 480 real-time PCR system (Roche Diagnostics K.K., Tokyo, Japan) according to the manufacturer's instructions. The PCR array experiment was performed in triplicate, and the gene expression levels were estimated using RT$^2$ Profiler™ PCR array data analysis software (Qiagen). Differences in expression levels between the 12-month-old and 6–7-week-old mice were calculated based on the difference in ⊿Ct values normalized to the levels of a housekeeping gene of heat shock protein 90-beta. $P$-values for differences in the expression levels between the 12-month-old and 6–7-week-old mice were assessed by Student's $t$-test. If the expression level showed greater than a twofold change or less than a 0.5-fold change, and was significantly different ($P<0.05$), the gene was considered upregulated (greater than a twofold change) or downregulated (less than a 0.5-fold change) in the 12-month-old compared with the 6–7-week-old mice.

## Gene specific real-time RT-PCR

The expression levels of nine inflammatory and immune-related genes were compared between the cochleae of 6–7-week-old (young control mice) and 3-, 6-, 9-, and 12-month-old mice by means of gene-specific real-time RT-PCR to investigate the time point when these inflammatory and immune-related gene expressions were modulated. The following nine immune-related genes were selected for the real-time RT-PCR analyses because our preliminary data, by means of next-generation sequencing (RNA-seq), suggested that these genes with important immune functions were upregulated in the cochleae of 12-month-old mice. In our preliminary data by RNA-seq, 800 genes were either upregulated (452 genes) or downregulated (348 genes) more than twofold in the aged cochleae of 12-month-old mice, compared with the cochleae of 6–7-week-old mice, and their functions were analyzed by bioinformatic analyses.

A cDNA library was synthesized by the reverse-transcription of the cochlear RNA samples (typically 500 ng) using the RT$^2$ First Strand Kit (Qiagen). Real-time PCR was performed using RT$^2$ SYBR Green qPCR Mastermix (Qiagen) and the specific primers for inflammatory and immune-related genes *Casp1* (PPM02921E; RT$^2$ qPCR Primer Assay, Qiagen), *IL18r1* (PPM03555B), *IL18rap* (PPM03137A), *IL1B* (PPM03109F), *Card9* (PPM40791A), *Clec4e* (PPM06261F), *Ifit1* (1PPM05530E), *Ifit3* (PPM06008B), and *Tlr9* (PPM04221A) using a thermal cycler (PTC-200; CFX Connect™, BioRad, Hercules, CA, USA). The gene expression level in each sample was calculated according to the ΔΔCt method with normalization to the level of the internal control, *Actb* (beta actin; PPM02945B).

The levels in 6–7-week-old and 3-, 6-, 9-, and 12-month-old cochleae were expressed as the mean ± standard deviation (SD) and compared using analysis of variance and the Bonferroni post-hoc test ($n$ = 6 for each time point, $P<0.05$). The specific primers for each gene were designed and experimentally verified for real-time PCR analyses by Qiagen. Information on the gene-specific primers is available on the Qiagen homepage (https://geneglobe.qiagen.com/product-groups/rt2-qpcr-primer-assays).

## Bioinformatic analyses of functions of the differentially expressed genes (DEGs)

Real-time RT-PCR array experiments generated a list of differentially expressed genes (DEGs), which were significantly upregulated (greater than a twofold change) or downregulated (less than a 0.5-fold change) in the cochleae of the 12-month-old mice as compared with the 6–7-week-old mice ($P<0.05$ by $t$-test).

First, the biological pathways associated with these DEGs were investigated using the Kyoto Encyclopedia of Genes and Genomics (KEGG) pathway analysis with the David Bioinformatics Resources 6.8 web-based genome database (https://david.ncifcrf.gov/) [13, 14]. The KEGG pathway is the annotation of functional gene pathways involving a group of genes. If a subset of DEGs is identified in abundance in a KEGG pathway with a $P$-value $<0.05$ and a false discovery rate (FDR) $<0.05$, these DEGs are considered as significantly enriching this pathway with a specific biological function.

Second, which transcription factors might regulate the expression of these DEGs was investigated to help understand how these DEGs are regulated by the upstream gene transcription mechanisms. Candidates for the transcription factors regulating these DEGs were identified based on predictions by the web-based Molecular Signatures Database 7.2 (MSigDB; https://www.gsea-msigdb.org/) [15, 16]. This analysis assessed the presence of DNA sequences targeted by transcription factors in the genomic regulatory region of the DEGs based on a $P$-value $<0.05$ and an FDR $<0.05$.

Third, the gene expression network of DEGs was studied using the web-based STRING 11.0 analysis tool (https://string-db.org) [17]. STRING is a program that analyzes the mutual relationships of proteins evidenced by their experimentally verified interactions, co-expressions, and co-citations in curated databases and PubMed abstracts, and visualizes the genes/ protein association networks. The list of the DEGs was subjected to STRING with a medium confidence score of 0.4.

## Immunohistochemistry

Immunohistochemical analysis of paraffin sections of cochleae from 12-month-old mice ($n = 3$) and 6–7-week-old mice ($n = 3$) was performed as previously described, with minor modifications [12]. After heat-mediated antigen retrieval, reactions were performed using anti–IL-18r1 antibody (ab231554; Abcam, Cambridge, UK, diluted 1/50), anti–IL-1B antibody (ab9722; Abcam, diluted 1/100), anti-TNFα antibody (ab 6671; Abcam, diluted 1/50), and anti-PTGS2 antibody (ab 15191; Abcam, diluted 1/100) at 4˚C overnight, followed by visualization using the ABC method (Vectastain Elite ABC Kit; Vector Laboratories, Burlingame, CA, USA). Rabbit polyclonal antibody raised against IBA1 (Ionized calcium binding adaptor protein 1), a macrophage/microglia-specific calcium-binding protein [18], was used to detect macrophages by immunohistochemistry (A3160; ABclonal, Tokyo, Japan, diluted 1/50). Immunofluorescent visualization of specific IL-18r1, IL-1B, TNFα and PTGS2 immunoreactivities was also performed using Alexa Fluor 568 donkey anti-rabbit IgG (A10042; Thermo Fisher Scientific, diluted 1/200) at 4˚C for 30 m. Tissue autofluorescence was eliminated using an autofluorescence quenching kit (TrueVIEW kit, Vector Laboratories) following the manufacturer's protocol, and nuclear counterstaining was performed using diamidino-2-phenylindole (DAPI). The specificity of the primary antibodies to IL-18r1, IL-1B, TNFα, and PTGS2 was verified by western blotting by the manufacturer (Abcam). As a negative control, sections were incubated with nonspecific rabbit IgG (5 µg/mL) and then visualized. Light and fluorescent microscopic images were acquired using a fluorescent microscope (BX-51-54; Olympus, Tokyo, Japan).

## Results

### Hearing levels in mice with ARHL

As shown in Fig 1, the c-ABR threshold was significantly higher in 6-month-old mice (63.0 ±14.9 dB SPL, mean±SD, $n = 10$) and 12-month-old mice (66.9±13.6 dB SPL, $n = 8$) than in 6–7-week-old mice (40±2.9, $n = 7$) ($P<0.01$), confirming that 6- and 12-month-old mice exhibited significant ARHL. The c-ABR thresholds in the 6- and 12-month-old mice were shifted by 23.0 and 26.9 dB, respectively, compared with the younger control mice.

### Real-time RT-PCR array and bioinformatic analyses of DEGs

A volcano plot (Fig 2) shows the gene expression profiles of 84 genes in the inflammatory and immune pathways analyzed by the real-time RT-PCR array. Each gene was plotted as a function of the ratio of expression levels between the 12-month-old and 6–7-week-old mice (the x-axis) and *P*-values for the differential expression levels between the two age groups (the y-axis). Among the 84 targeted genes, as many as 40 were significantly upregulated (33 genes) or downregulated (7 genes) with greater than a twofold change in the cochleae of 12-month-old compared with 6–7-week-old mice ($P<0.05$). Table 1 summarizes the gene symbols and names of these 40 DEGs from among the 84 analyzed immune-related genes.

These DEGs were shown to play roles in the biological functions of the top five significant KEGG functional pathways abundantly enriched by the DEGs, including cytokine–cytokine

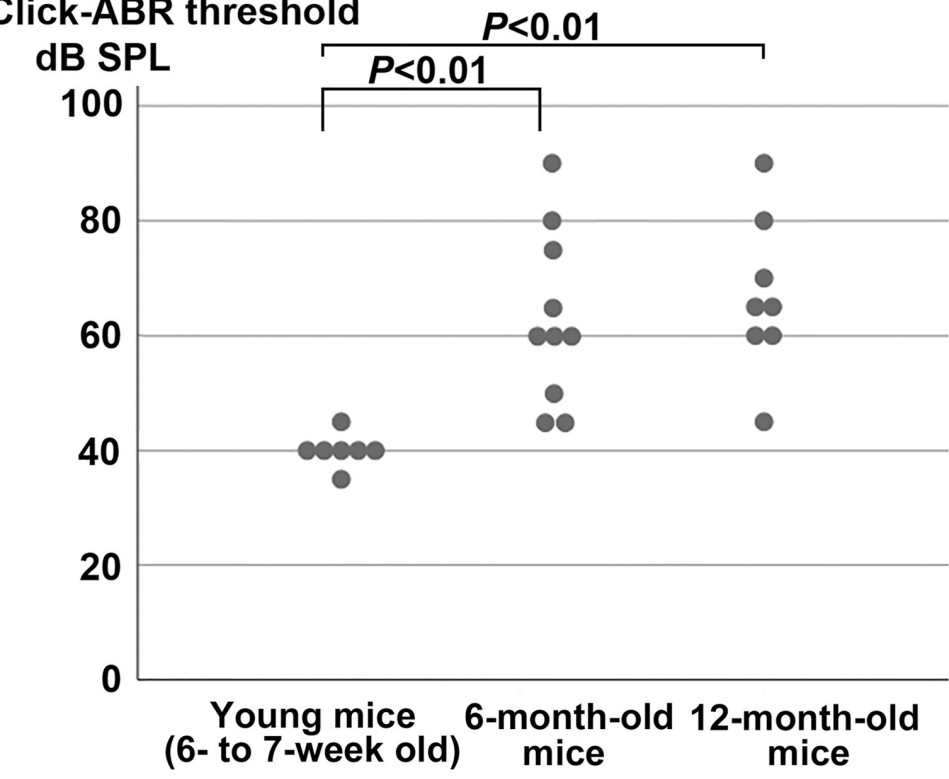

**Fig 1. Click-auditory brainstem response (c-ABR) thresholds demonstrating age-related hearing loss in male C57BL/6J mice.** The mean ± standard deviation c-ABR threshold was significantly higher in 6-month-old mice (63.0 ±14.9 dB sound pressure level, *n* = 10) and 12-month-old mice (66.9±13.6, *n* = 8) than in 6–7-week-old mice (40±2.9, *n* = 7) (*P*<0.01, Kruskal–Wallis and Mann–Whitney *U* tests).

receptor interaction (involving 29 DEGs; *P* = 6.4E-35; FDR = 5.9E-33), the chemokine signaling pathway (18 DEGs; *P* = 7.0E-18; FDR = 3.2E-16), the TNF signaling pathway (13 DEGs; *P* = 7.0E-14; FDR = 2.2E-12), and the Toll-like receptor signaling pathway (11 DEGs; *P* = 3.4E-11; FDR = 6.2E-10) (Table 2).

The analysis of transcription factor targets in the MSigDB database revealed that a transcription factor, nuclear factor-kappa B (NF-κB), was predicted to interact *in-trans* with the genomic regulatory sequences of eight upregulated DEGs: *Ccl5* (chemokine (C-C motif) ligand 5), *Cxcl10* (chemokine (C-X-C motif) ligand 10), *Cxcl9* (chemokine (C-X-C motif) ligand 9), *Il1a* (IL 1 alpha), *Lta* (lymphotoxin A), *Ltb* (lymphotoxin B), *Ptgs2* (prostaglandin-endoperoxide synthase 2), and *Tnf* (tumor necrosis factor), with a *P*-value <0.01 and an FDR <0.01.

Based on the DNA sequence information deposited in the database, the genomic regulatory regions of these eight DEGs contain at least one NF-κB target sequence in the regions spanning up to 4 kb around the transcription starting site of each DEG. Furthermore, a subunit of the NF-κB transcription factor complex, *Nfkb1* (nuclear factor of kappa light polypeptide gene enhancer in B-cells 1, p105), was significantly upregulated in the cochleae of aged compared with young mice in our dataset for the real-time RT-PCR array.

Fig 3, which was created using the STRING program, shows the protein–protein association network of the immune-related DEGs actively controlled in aged cochleae. The figure illustrates that the transcription factor *Nfkb1* is upregulated in aged cochleae, and in turn, NF-

## Cochlear gene expressions in inflammatory and immune pathways (84 genes)

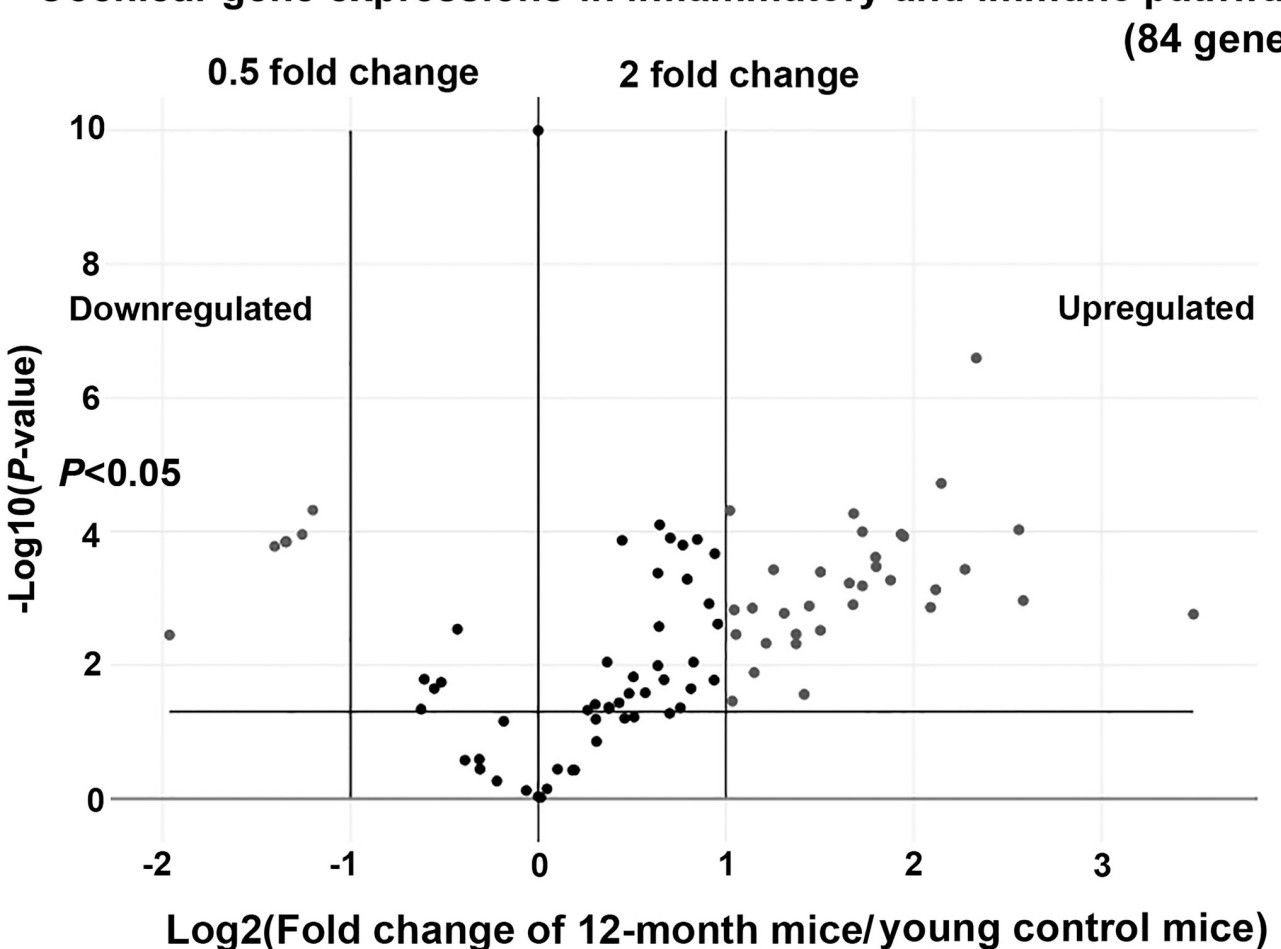

**Fig 2. A volcano plot showing the ratios of the gene expression levels in the cochleae of 12-month-old mice relative to young control mice.** Data of 84 key genes involved in inflammatory and immune functions are plotted. Forty of the 84 genes are differentially expressed with greater than twofold upregulation (greater than a twofold change, 33 genes) or downregulation (less than a 0.5-fold change, 7 genes), as well as significant P-values to indicate differential expression levels (P<0.05, t-test) between the cochleae of 12-month-old vs. 6–7-week-old mice (young control mice). The x-axis indicates Log2 (fold change of 12-month-old mice/young control mice) showing the relative expression levels. The y-axis represents–Log10 (P-value) to assess significant differences between the two groups (n = 3 for each gene in the 12-month-old and young control mice).

κB binds to the genomic regulatory regions of eight upregulated DEGs—*Ccl5*, *Cxcl10*, *Cxcl9*, *Il1a*, *Lta*, *Ltb*, *Ptgs2*, and *Tnf*—which are mutually associated in the network of the 40 immune-related DEGs.

### Gene-specific real-time RT-PCR

Fig 4 summarizes the respective mean±SD expression levels of the nine inflammatory and immunity-related genes at each time point: *Casp1* (6–7-week-old mice: 1.01±0.07 -fold change; 3-month-old mice: 1.00±0.05; 6-month-old mice: 0.97±0.04; 9-month-old mice: 1.00±0.06; 12-month-old mice: 2.88±0.16), *IL18r1* (1.00±0.07, 1.20±0.06, 1.00±0.05, 1.17±0.06, 1.49 ±0.08), *IL18rap* (1.01±0.12, 0.99±0.15, 1.41±0.25, 1.41±0.12, 1.62±0.07), *IL1B* (1.00±0.07, 1.74 ±0.12, 2.02±0.14, 1.58±0.06, 3.51±0.29), *Card9* (1.00±0.07, 1.19±0.04, 1.36±0.10, 1.08±0.06, 2.58±0.10), *Clec4e* (1.01±0.10, 1.24±0.12, 1.39±0.15, 1.23±0.15, 3.39±0.32), *Ifit1* (1.01±0.11, 1.68±0.16, 1.20±0.10, 1.11±0.07, 5.49±0.52), *Ifit3* (1.00±0.09, 1.92±0.18, 2.14±0.17, 1.66±0.15,

**Table 1. Inflammatory and immune-related genes upregulated or downregulated in the cochleae of 12-month-old compared with 6–7-week-old mice.**

| Gene symbol | Gene name | P-value (*t*-test) | Fold change |
|---|---|---|---|
| **Upregulated genes** | | | |
| Ccl12 | Chemokine (C-C motif) ligand 12 | 0.00034 | 3.48 |
| Ccl2 | Chemokine (C-C motif) ligand 2 | 0.00065 | 3.31 |
| Ccl5 | Chemokine (C-C motif) ligand 5 | 0.00124 | 3.2 |
| Ccl7 | Chemokine (C-C motif) ligand 7 | 0.00037 | 2.38 |
| Ccl8 | Chemokine (C-C motif) ligand 8 | 0.00173 | 11.24 |
| Ccr1 | Chemokine (C-C motif) receptor 1 | 0.00130 | 2.72 |
| Ccr2 | Chemokine (C-C motif) receptor 2 | 0.00005 | 3.2 |
| Ccr3 | Chemokine (C-C motif) receptor 3 | 0.00040 | 2.83 |
| Ccr7 | Chemokine (C-C motif) receptor 7 | 0.00010 | 3.31 |
| Cxcl10 | Chemokine (C-X-C motif) ligand 10 | 0.00012 | 3.85 |
| Cxcl9 | Chemokine (C-X-C motif) ligand 9 | 0.00009 | 5.9 |
| Cxcr1 | Chemokine (C-X-C motif) receptor 1 | 0.00037 | 4.83 |
| Cxcr2 | Chemokine (C-X-C motif) receptor 2 | 0.00059 | 3.15 |
| Cxcr4 | Chemokine (C-X-C motif) receptor 4 | 0.00024 | 3.47 |
| Fasl | Fas ligand (TNF superfamily, member 6) | 0.02733 | 2.67 |
| Ifng | Interferon gamma | 0.01286 | 2.22 |
| Il1a | Interleukin 1 alpha | 0.00470 | 2.32 |
| Il1b | Interleukin 1 beta | 0.00053 | 3.67 |
| Il6 | Interleukin 6 | 0.00002 | 4.43 |
| Il7 | Interleukin 7 | 0.00342 | 2.59 |
| Kng1 | Kininogen 1 | 0.00136 | 4.26 |
| Lta | Lymphotoxin A | 0.00301 | 2.83 |
| Ltb | Lymphotoxin B | 0.00140 | 2.2 |
| Nfkb1 | Nuclear factor of kappa light polypeptide gene enhancer in B-cells 1, p105 | 0.00005 | 2.03 |
| Nr3c1 | Nuclear receptor subfamily 3, group C, member 1 | 0.00346 | 2.08 |
| Ptgs2 | Prostaglandin-endoperoxide synthase 2 | 0.00477 | 2.59 |
| Sele | Selectin, endothelial cell | 0.03442 | 2.05 |
| Tlr1 | Toll-like receptor 1 | 0.00074 | 4.34 |
| Tlr6 | Toll-like receptor 6 | 0.00108 | 5.99 |
| Tlr7 | Toll-like receptor 7 | 0.00000 | 5.04 |
| Tlr9 | Toll-like receptor 9 | 0.00011 | 3.82 |
| Tnf | Tumor necrosis factor | 0.00149 | 2.06 |
| Tnfsf14 | Tumor necrosis factor (ligand) superfamily, member 14 | 0.00167 | 2.48 |
| **Downregulated genes** | | | |
| Ccl1 | Chemokine (C-C motif) ligand 1 | 0.00014 | 0.39 |
| Ccl20 | Chemokine (C-C motif) ligand 20 | 0.00014 | 0.39 |
| Crp | C-reactive protein, pentraxin-related | 0.00005 | 0.43 |
| Cxcl3 | Chemokine (C-X-C motif) ligand 3 | 0.00352 | 0.26 |
| Il17a | Interleukin 17A | 0.00017 | 0.38 |
| Il22 | Interleukin 22 | 0.00014 | 0.39 |
| Il9 | Interleukin 9 | 0.00011 | 0.42 |

The expressions of 84 immune-related genes were analyzed by real-time RT-PCR array. As a result, 33 upregulated and 7 downregulated genes were detected with greater than a twofold or less than a 0.5-fold change, respectively, with a *P*-value <0.05 for significantly different expression levels in the 12-month-old relative to the 6–7-week-old mice (*n* = 3, *t*-test).

**Table 2. Functional gene pathways associated with the differentially expressed genes (DEGs) in 12-month-old relative to 6–7-week-old cochleae (top 5 significant pathways).**

|  | Gene count | *P*-value(<0.05) | FDR(<0.05) |
|---|---|---|---|
| Cytokine-cytokine receptor interaction | 29 | 6.4E-35 | 5.9E-33 |
| Chemokine signaling pathway | 18 | 7.0E-18 | 3.2E-16 |
| TNF signaling pathway | 13 | 7.0E-14 | 2.2E-12 |
| Rheumatoid arthritis | 11 | 4.0E-12 | 9.2E-11 |
| Toll-like receptor signaling pathway | 11 | 3.4E-11 | 6.2E-10 |

The biological functions of 40 DEGs identified by the real-time RT-PCR array were significantly associated with these top five KEGG pathways with a *P*-value <0.01 and a false discovery rate (FDR) <0.01.

1.75±0.14), and *Tlr9* (1.00±0.09, 0.95±0.17, 0.71±0.05, 1.40±0.09, 3.26±0.33). As shown in the figure, the major proinflammatory molecule *IL1B* was significantly upregulated by more than a 1.5-fold change at the early time point of 3 months and thereafter, as compared with the level at 6–7 weeks. *IL18rap* was significantly upregulated by more than a 1.4-fold change at the time point of 6 months and thereafter. An interferon-induced gene, *Ifit3*, was also upregulated by more than a 1.5-fold change at the time point of 3 months and thereafter. Seven out of nine inflammatory and immunity-related genes (except for *IL18rap* and *Ifit3*) examined by gene-specific real-time RT-PCR showed significant and noticeable upregulation between the late time points of 9 and 12 months ($P<0.01$, $n = 6$ for each time point).

## Immunohistochemistry

The IL-18r1 receptor and IL-1B were found to be upregulated in 12-month-old cochleae in the real-time RT-PCR analyses. Immunohistochemical analysis of cochlear histochemical sections from 12-month-old mice demonstrated that both proteins localized in the spiral ligament, spiral limbus, and organ of Corti (upper insets in Fig 5). Because the organ of Corti in 12-month-old C57BL/6 mice showed signs of age-related degeneration in the basal turn of the cochleae in previous studies [19], the organ of Corti in the apical turn was examined in our immunohistochemical analysis in the aged mice. In the cochleae of 6-week-old mice, IL-18r1 and IL-1B localized in the spiral ligament, the spiral limbus, and the organ of Corti (lower insets in Fig 5). Cochlear localization of IL-18r1 and IL-1B was similar between the aged and younger mice.

Immunohistochemical analysis of the two NF-κB-interacting inflammatory molecules (TNFα and PTGS2) showed that they were expressed ubiquitously in the cochlear structures of the 12-month-old mice. Unequivocal immunoreactivity to TNFα and PTGS2 was observed in the lateral wall (the spiral ligament and stria vascularis) of the aged cochleae (upper insets in Fig 6). Cochlear localization of TNFα and PTGS2 in the 6-week-old mice was similar to that in the 12-month-old mice (lower insets in Fig 6).

As shown in Fig 7, IBA1-positive macrophages were observed in the stria vascularis and the inferior division of the spiral limbus in the cochleae of 12-month-old mice. No IBA1-positive cells were found in the cochlear structures of 6-week-old mice.

## Discussion

In this study, we hypothesized that inflammation and immune reactions are regulated during the aging process of mice cochlear tissues with ARHL. To test this hypothesis, we analyzed the expression levels of 84 key genes known to be involved in inflammatory and immune functions in aged and young cochleae. Of the 84 genes examined by a real-time RT-PCR array, the expressions of 40 (47.6%) were differentially regulated in the cochleae of 12-month-old

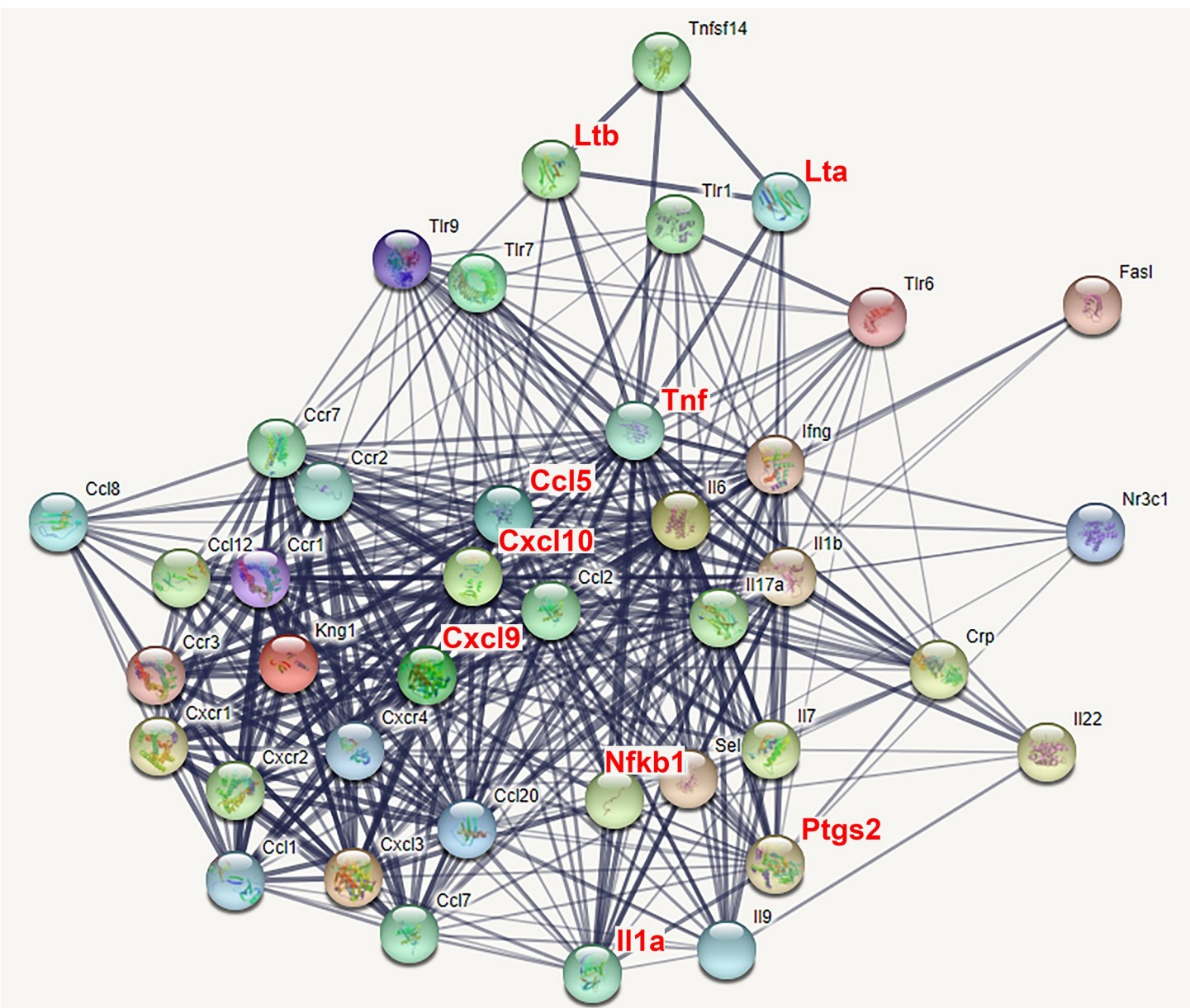

**Fig 3. Expression network of 40 differentially regulated genes (DEGs) with inflammatory and immune functions in 12-month-old cochleae.** A web-based STRING database (https://string-db.org) computed a graphical representation of the mutual relationships of the 40 immune-related DEGs detected by the real-time RT-PCR array, based on their experimentally verified interactions, co-expressions, and co-citations in curated databases and PubMed abstracts. The figure shows that a transcription factor, *Nfkb1*, is upregulated in aged cochleae, and that NF-κB interacts with the genomic regulatory regions of the eight upregulated DEGs (*Ccl5*, *Cxcl10*, *Cxcl9*, *Il1a*, *Lta*, *Ltb*, *Ptgs2*, and *Tnf*) mutually associated in the expression network of the 40 immune-related DEGs in aged cochleae.

compared with 6–7-week-old mice, 33 of which were upregulated in the aged cochleae. The results of our experiments supported the hypothesis that inflammatory and immune reactions are modulated in aged cochlear tissues. The differential expressions of these immune-related genes were involved in functional gene pathways such as cytokine–cytokine receptor interaction, the chemokine signaling pathway, the TNF signaling pathway, and the Toll-like receptor signaling pathway.

A transcription factor, NF-κB, was upregulated in the aged cochlear tissue and predicted to bind to the genomic regulatory sequences of eight upregulated DEGs: *Ccl5*, *Cxcl10*, *Cxcl9*, *Il1a*,

# Expression levels of immune-related genes in cochleae

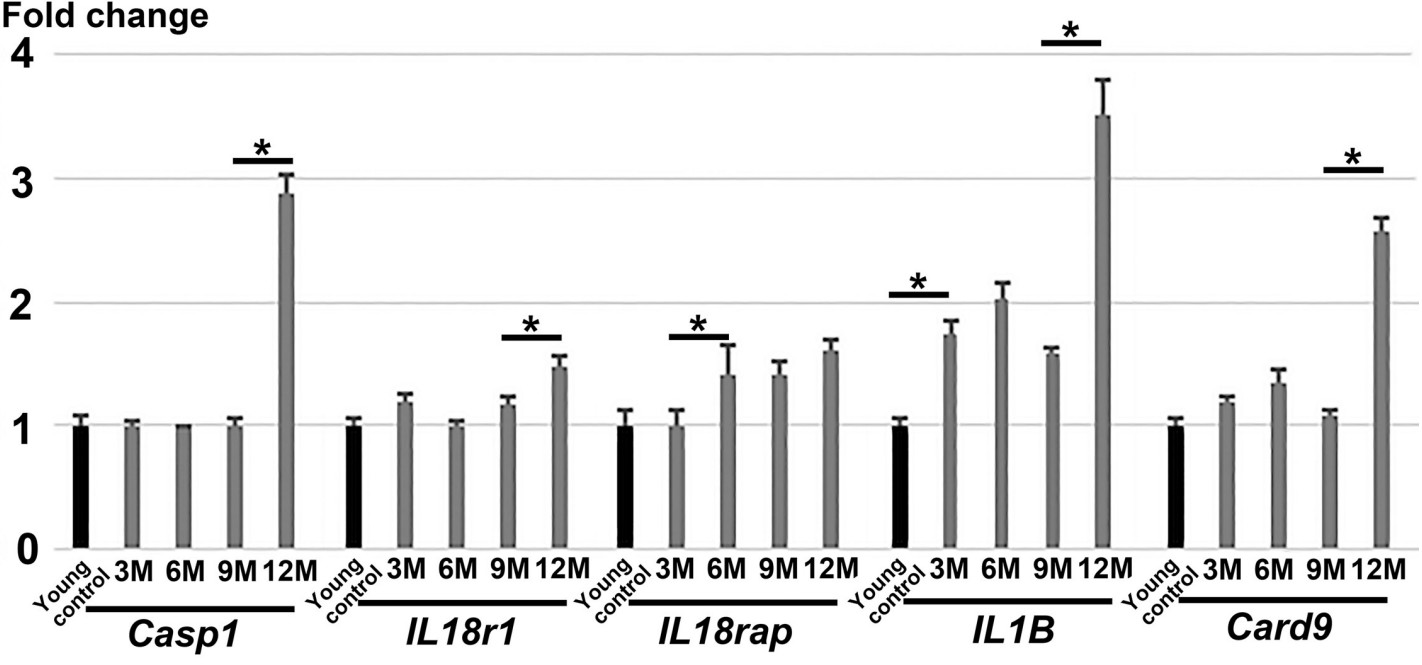

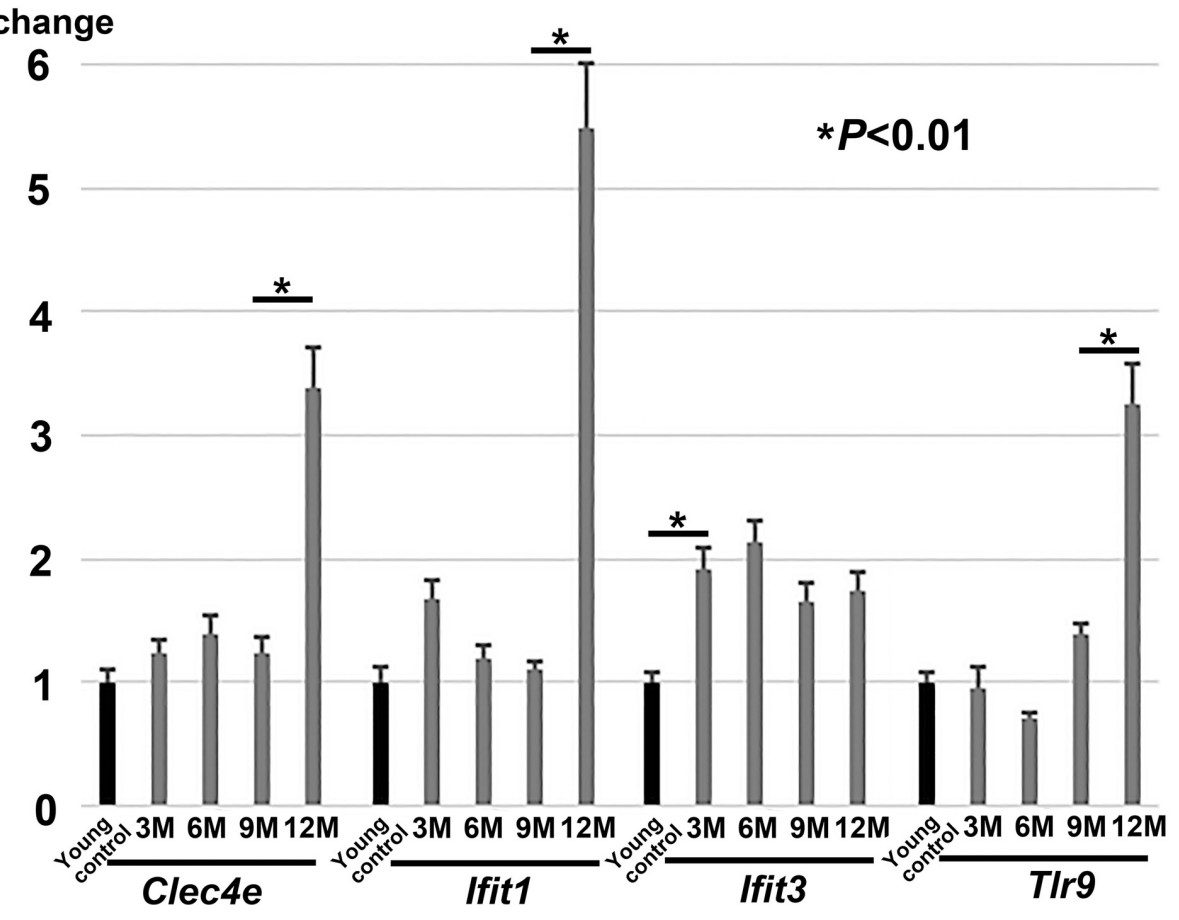

**Fig 4. Gene-specific real-time RT-PCR of immune-related genes in the cochleae of 6–7-week-old (young control) and 3-month-old (3M), 6-month-old (6M), 9-month-old (9M), and 12-month-old (12M) mice.** Gene expression levels are analyzed for the nine inflammatory and immune-related genes—*Casp1*, *IL18r1*, *Il18rap*, *IL1B*, *Card9*, *Clec4e*, *Ifit1*, *Ifit3*, and *Tlr9*—at each age of the mice cochleae. The expressions of the major proinflammatory molecules *IL1B* and *IL18rap* are upregulated by 3M and 6M, respectively, and thereafter during the aging process as compared with the levels in the young control cochleae. *Ifit3* is also upregulated by 3M and thereafter. The expressions of seven of the nine immune-related genes (except for *IL18rap* and *Ifit3*) show significant upregulation between the ages of 9M and 12M (*n* = 6 for each time point, *P*<0.01 by analysis of variance and the Bonferroni post-hoc test).

*Lta*, *Ltb*, *Ptgs2*, and *Tnf*. In general, NF-κB is a master regulator controlling gene expressions pertaining to innate immunity and plays a role in the aging process [20]. In cochleae during the pathological processes of noise-induced hearing loss and ARHL, NF-κB transcriptional activity was strongly induced in the spiral ligament and stria vascularis of the lateral wall [21]. In the immunohistochemical analyses in the present study, two NF-κB-interacting molecules, TNFα and PTGS2, were also expressed in the spiral ligament and stria vascularis of the lateral wall in aged cochleae. Our gene expression analyses corroborated the data that NF-κB may control the transcriptional network of immune-related genes during the aging process of the cochlea.

In agreement with our data, recent study by Su et al. showed that the gene expressions involved in inflammatory and immune functions were upregulated in the cochleae of 12-month-old C57BL/6J mice as compared with 4-week-old mice by means of next-generation sequencing (RNA-seq) [10]. However, they did not provide information on the age of mice when the inflammatory and immunity-related gene expressions were modulated in the cochleae during the aging process. Our data, obtained by gene-specific real-time RT-PCR, provide new evidence that the major proinflammatory molecules *IL1B* and *IL18rap* are significantly upregulated in cochleae at 3 and 6 months, respectively, and thereafter in aging cochleae. Upregulation of an interferon-induced gene, *Ifit3*, was also observed at 3 months and thereafter. Subsequently during the aging process, seven of the nine immune-related genes examined in our experiments (*Casp1*, *IL18r1*, *IL1B*, *Card9*, *Clec4e*, *Ifit1*, and *Tlr9*) showed significant upregulation between the late time points of 9 and 12 months. According to a previously published paper [11], ABR thresholds in the high-frequency range (32 kHz) in inbred C57BL/6J mice were 45.0 dB SPL at 3 months of age, and subsequently showed threshold shifts of 33.1, 38.4, and 47.1 dB at 6, 9, and 12 months, respectively, compared with the threshold at 3 months. Based on these observations, *IL1B*, *IL18rap*, and *Ifit3* may play significant roles in the development of ARHL by 6 months of age. *IL1B* and the six other genes, *Casp1*, *IL18r1*, *Card9*, *Clec4e*, *Ifit1*, and *Tlr9*, may participate in the immune response to the degeneration of the cochlear cells because C57BL/6J mice exhibited significant ARHL as early as 6 months of age [11]. These data suggest that a remarkable progression of the innate immunity process occurred in mice cochleae at an advanced stage of ARHL (between 9 and 12 months of age).

By means of gene-specific RT-PCR, Shi et al. demonstrated upregulation of NOD-like receptor family pyrin domain containing 3 (NLRP3) inflammasome genes (*Casp1*, *IL18*, and *IL1B*) in the cochleae of 12-month-old mice [9]. These NLRP3 inflammasome molecules, which were also examined in our gene-specific real-time RT-PCR, comprise a key innate immune pathway involved in the recognition of molecular triggers that appear during cellular senescence [22]. Among the genes analyzed in our experiments, *Card9* and *Clec4e* activated macrophage inflammatory responses, which may play roles in chronic inflammatory diseases [23, 24]. Both *Ifit1 and Ifit3* are interferon-inducible genes that control pro-inflammatory gene programming in macrophages [25, 26]. *Tlr9* recognizes its ligands of pathogenic DNA in immune cells and triggers signaling cascades that lead to the induction of type 1 interferon expression and pro-inflammatory cytokine responses [27].

The downregulation of seven immune-related genes—*Ccl1*, *Ccl20*, *Cxcl3*, *IL17a*, *IL22*, *IL9*, and *Crp*—was demonstrated by the RT-PCR array analysis in 12-month-old cochleae. The

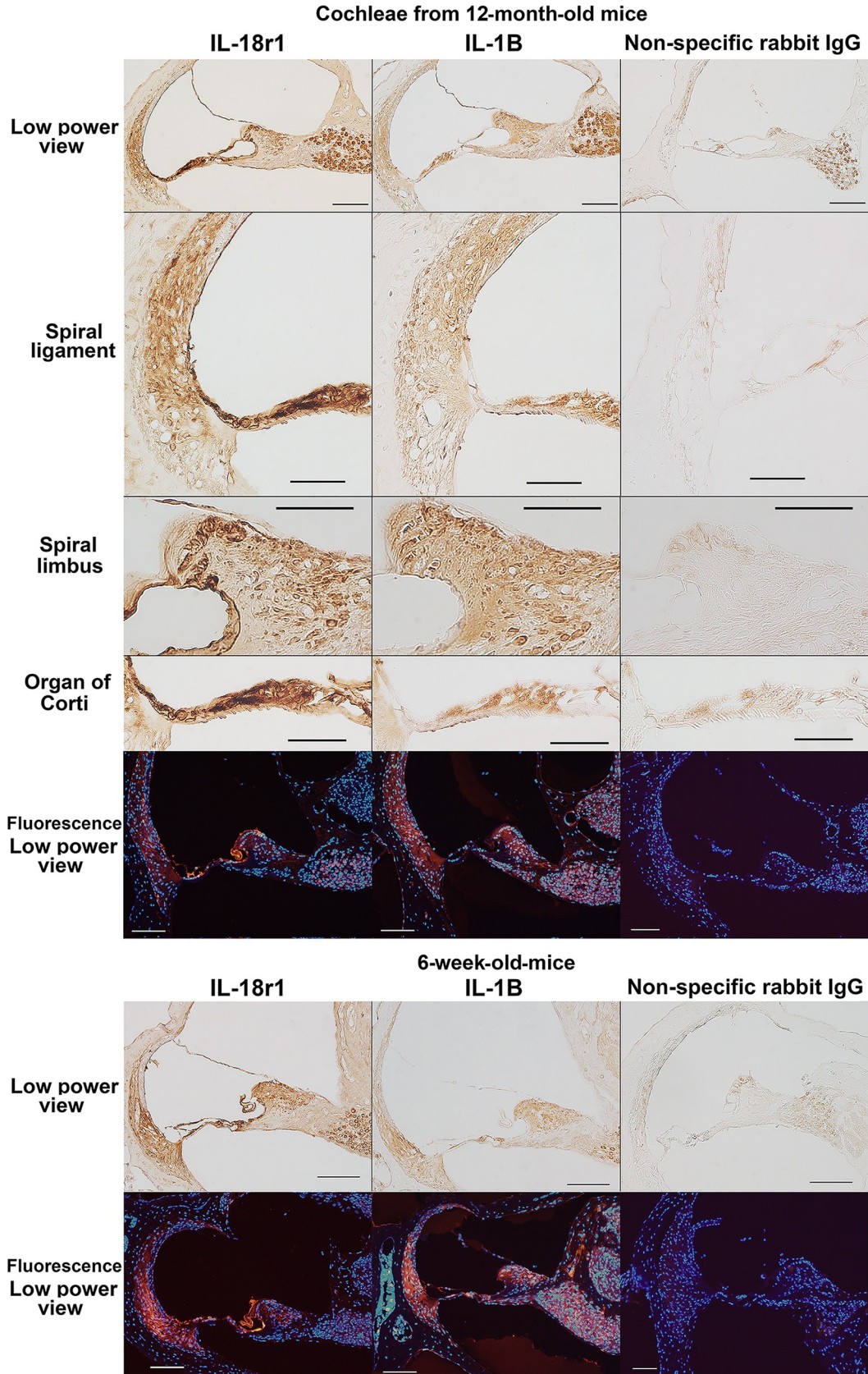

**Fig 5. Immunohistochemical analysis of IL-18 receptor 1 (IL-18r1) and IL-1 beta (IL-1B) expression in aged cochleae.**
The figures show immunoreactivity to IL-18r1 and IL-1B in the cochleae of 12-month-old and 6-week-old mice. Negative control sections incubated with nonspecific rabbit IgG showed no signals except for background staining in the spiral neurons in sections visualized using the ABC method. Therefore, the signals in the spiral neurons may be due to the nonspecific protein binding of rabbit IgG. In the immunofluorescent figures, blue and red indicate nuclear staining by DAPI and immunoreactivity to IL-18r1 and IL-1B, respectively. Scale bars indicate 100 μm (low-power view) and 50 μm (insets showing the spiral ligament, spiral limbus, and organ of Corti).

three downregulated chemokine-ligand genes—*Ccl1*, *Ccl20*, and *Cxcl3*—were closely related to each other in the gene expression network of the 40 DEGs shown in Fig 3. The three interleukins—*IL17*, *IL22*, and *IL9*—were also closely associated in the gene expression network. *Ccl1* and *IL9* are known to be involved in the anti-apoptotic activity of immune cells [28, 29].

In our immunohistochemical analyses, IL-18r1 and IL-1B proteins localized to the spiral ligament, spiral limbus, and organ of Corti in cochlear sections from aged mice. IL-18 and IL-1B are major proinflammatory molecules that participate in the inflammasome pathway, and RT-PCR analyses have shown that their encoding genes are upregulated in aged cochleae. In a previous report, the expression of an inflammasome-forming protein, NLRP3 was detected by immunohistochemistry in cochlear structures, including the spiral neurons, in aged mice with ARHL [9]. These data therefore suggest that innate immune reactions play significant roles in the aging process in these cochlear structures. In addition, IBA1-positive macrophages were observed in the stria vascularis and the inferior division of the spiral ligament. These lateral wall structures (the stria vascularis and the inferior division of the spiral ligament) are thought to be the primary anatomical structure of leukocyte migration into the cochleae because of their abundantly dense vasculature [30].

Chronic low-grade inflammation plays a key role in age-related diseases such as Alzheimer's disease via a process called inflammaging [31]. An epidemiologic study of 611 older adults in the United Kingdom showed that increases in serum inflammatory markers, including white blood cell count, neutrophil count, IL-6 levels, and C-reactive protein levels, were significantly associated with worse hearing levels, as demonstrated statistically by multiple regression models after adjusting for the covariates of age, gender, smoking status, and exposure to noise at work [7]. Based on such data, a clinical trial was conducted to investigate whether continuous oral administration of low-dose aspirin prevented or reduced ARHL in a 3-year study [32]. In animal experiments aimed at developing immunologic ARHL therapies, hearing loss, degeneration of the spiral neurons, and T-cell dysfunction observed in 6-month-old mice recovered in 12-month-old mice that had received two fetal thymus transplants [33].

A recent study by Srivastava et al. analyzed changes in the transcriptome using RNA-seq in the brain, blood, skin, and liver of C57BL/6 mice at 9, 15, 24, and 30 months of age [34]. They found that the most significant DEGs in the aged brain, blood, and liver were upregulated genes of inflammation and immune function. Compared with the brain, blood, and liver, only a few genes of inflammation and immune function were differentially regulated in the aged skin. We therefore speculate that DEGs involved in the immune system may be a common characteristic found in the transcriptome of aged nervous systems, including the auditory system. Age-related changes in the gene expressions unique to the cochlea might also be present because of its direct exposure to environmental stress (noise).

The gene and protein expression data in this study showed that inflammatory and immune reactions were modulated in aged cochlear tissues with ARHL. How these inflammatory and immune reactions positively or negatively impact the pathologic mechanisms of ARHL should be further clarified in animal experiments. These data could serve as the basis for the development of preventive and therapeutic measures for treating ARHL by targeting immune function in the cochleae.

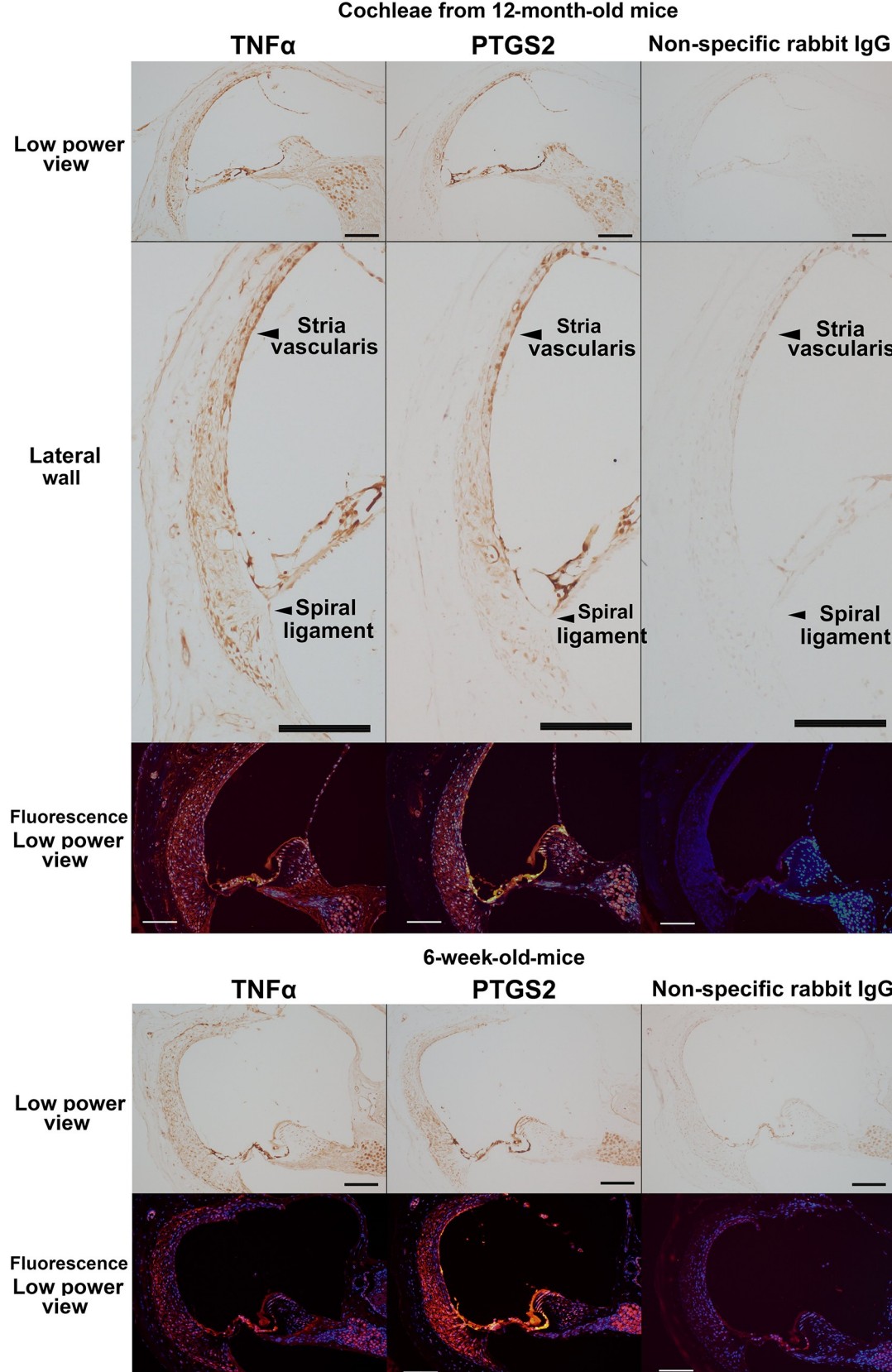

**Fig 6. Immunohistochemical analysis of tumor necrosis factor α (TNFα) and prostaglandin-endoperoxide synthase 2 (PTGS2) expression in aged cochleae.** The figures show immunoreactivity to TNFα and PTGS2 in the cochleae of 12-month-old and 6-week-old mice. In the immunofluorescent figures, blue and red indicate nuclear staining by DAPI and immunoreactivity to TNFα and PTGS2, respectively. Scale bars indicate 100 μm.

A limitation of this study is that the expression levels of the NF-κB-interacting upregulated genes (*Ccl5*, *Cxcl10*, *Cxcl9*, *Il1a*, *Lta*, *Ltb*, *Ptgs2*, and *Tnf*) were not analyzed by real-time RT-PCR at different ages (3-, 6-, 9-, and 12-month-old mice). Such data would provide important insights into the involvement of the NF-κB-interacting immune gene network in the molecular process of age-related hearing loss.

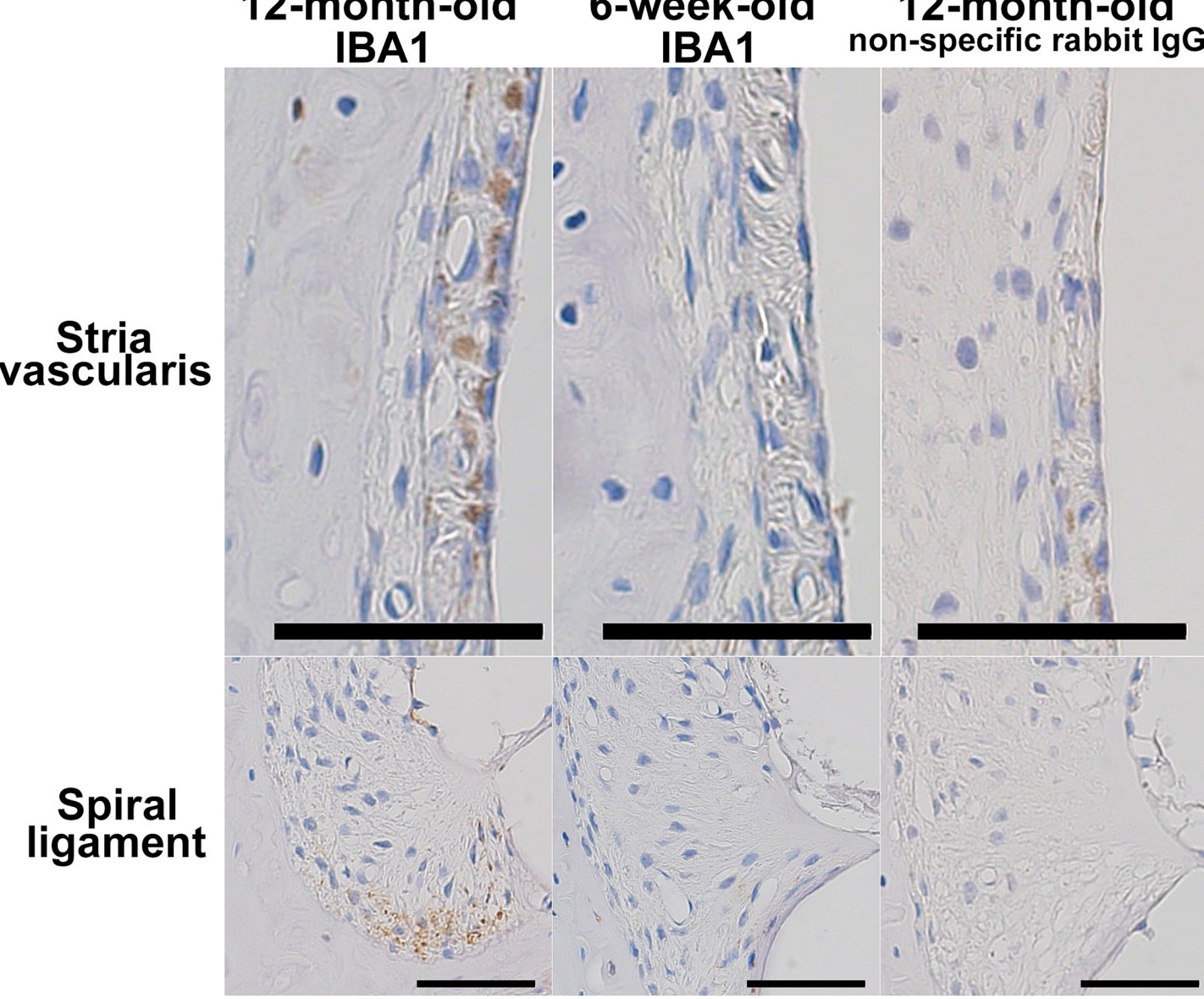

**Fig 7. Immunohistochemistry showing IBA1-positive macrophages in aged cochleae.** The figures show immunohistochemical detection of IBA1-positive macrophages in the stria vascularis and the inferior division of the spiral ligament in the cochleae of 12-month-old mice. No IBA1-positive cells were observed in the cochlear structures of 6-week-old mice. Blue and brown indicate nuclear staining by hematoxylin and immunoreactivity to IBA1, respectively. Negative control sections incubated with nonspecific rabbit IgG showed no signals. Scale bars indicate 50 μm.

## Supporting information

**S1 Table. Gene symbols, names, and cochlear expressions of the 84 immune-related genes examined by real-time RT-PCR array in this study.** The expressions of inflammatory and immune-related genes in the cochleae of 12-month-old relative to 6–7-week-old mice are tabulated. The expressions of 84 immune-related genes were analyzed by real-time RT-PCR array. As a result, 33 upregulated and 7 downregulated genes were detected with greater than a twofold or less than a 0.5-fold change, respectively, with a $P$-value <0.05 for significantly different expression levels between the 12-month-old and 6–7-week-old mice ($n = 3$, $t$-test). (XLSX)

## Author Contributions

**Conceptualization:** Kensuke Uraguchi, Yukihide Maeda, Shohei Fujimoto, Shin Kariya.

**Data curation:** Yukihide Maeda.

**Formal analysis:** Yukihide Maeda.

**Funding acquisition:** Yukihide Maeda, Shohei Fujimoto, Shin Kariya.

**Investigation:** Kensuke Uraguchi, Yukihide Maeda, Junko Takahara, Ryotaro Omichi, Shohei Fujimoto.

**Resources:** Kazunori Nishizaki, Mizuo Ando.

**Supervision:** Yukihide Maeda, Kazunori Nishizaki, Mizuo Ando.

**Writing – original draft:** Kensuke Uraguchi, Yukihide Maeda.

**Writing – review & editing:** Yukihide Maeda, Mizuo Ando.

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
