## [Decision Letter · Decision Letter 0]

14 Jul 2021

PONE-D-21-13350

Upregulation of a nuclear factor-kappa B-interacting immune gene network in mice cochleae with age-related hearing loss

PLOS ONE

Dear Dr. Maeda,

Thank you for submitting your manuscript to PLOS ONE. After careful consideration, we feel that it has merit but does not fully meet PLOS ONE’s publication criteria as it currently stands. Therefore, we invite you to submit a revised version of the manuscript that addresses the points raised during the review process.

We look forward to receiving your revised manuscript.

Kind regards,

Vasu Punj

Academic Editor

PLOS ONE

Journal Requirements:

2. We note that Figure 5 in your submission contain copyrighted images. All PLOS content is published under the Creative Commons Attribution License (CC BY 4.0), which means that the manuscript, images, and Supporting Information files will be freely available online, and any third party is permitted to access, download, copy, distribute, and use these materials in any way, even commercially, with proper attribution. For more information, see our copyright guidelines: http://journals.plos.org/plosone/s/licenses-and-copyright.

a. You may seek permission from the original copyright holder of Figure 5 to publish the content specifically under the CC BY 4.0 license. 

Reviewers' comments:

Reviewer's Responses to Questions

**Comments to the Author**

1. Is the manuscript technically sound, and do the data support the conclusions?

Reviewer #1: Yes

Reviewer #2: Yes

2. Has the statistical analysis been performed appropriately and rigorously? 

Reviewer #1: Yes

Reviewer #2: Yes

3. Have the authors made all data underlying the findings in their manuscript fully available?

Reviewer #1: Yes

Reviewer #2: Yes

4. Is the manuscript presented in an intelligible fashion and written in standard English?

Reviewer #1: Yes

Reviewer #2: Yes

5. Review Comments to the Author

Reviewer #1: In this manuscript, the authors have compared the expression of immune-related genes between young and old mice with manifested hearing loss and identified differentially expressed genes (DEGs), which expression could be associated with age-related hearing loss (ARHL). Among the significantly upregulated DEGs, eight genes (Ccl5, Cxcl10, Cxcl9, Il1a, Lta, Ltb, Ptgs2, and Tnf) were proposed transcriptionally regulated by NF-kB, which Nfkb1 subunit is also upregulated in aged cochleae. In addition, seven other immune-related genes (Casp1, IL18r1, IL1B, Card9, Clec4e, Ifit1, and Tlr9) were significantly upregulated at the advanced stage ARHL.

Although there is another recently published similar study (Su Z, et al. 2020, PeerJ. 2020;8:e9737. PubMed PMID: 32879802) analyzing the expression of a much higher number of the genes (731 genes) and showing the association of cochlear inflammation with ARHL, this study also represents a valuable contribution in clarification the role of inflammaging in development of ARHL.

Although the results are compelling and very relevant, there is still room for improvement.

Major issues

1. Keeping that authors showed significant upregulation of NF-kB in the aged cochlear tissue and predicted its involvement in transcriptional regulation of eight significantly upregulated DEGs (Ccl5, Cxcl10, Cxcl9, Il1a, Lta, Ltb, Ptgs2, and Tnf), it is not clear why they did not show the expression of these genes on different ages (3-month-old, 6-month-old, 9-month-old and 12-month-old mice) as it was shown for nine genes selected based on the preliminary data from next-generation sequencing (RNAseq).

2. Although the authors referred in the discussion part that the hearing status is getting to worsen as animals are older, they should assess the hearing status of animals at all analyzed ages (at least at 7-weak, 6- month and 12- month age) in parallel with the gene expression analysis. That will help understand whether the up-regulation of NF-kB-regulated genes is associated with the development of AHR or is only manifested in the oldest animals (12-month age).

3. Immunohistochemical analysis of cochlear sections from mice at different ages should be also extended for some of NF-kB -regulated significantly upregulated DEGs, as it was done for IL-18r1 and IL-1B.

4. Considering the role of macrophages in the cochlear inflammatory response and the immune capacity of cochlear supporting cells, co-staining inflammatory mediators and cell-specific markers will contribute to identifying the primary cellular source of up-regulated cytokines and immune-related factors associated with ARHL.

Minor issues

1. Although authors called on their preliminary data, they should clarify how many genes were analyzed by next-generation sequencing (RNAseq). This statement will also put weight on the paper, considering that a similar study analyzing more than 700 genes has already been published.

2. There is a terminology issue that should be solved.

a) Interleukins and chemokines are cytokines as well, and they should not be listed separately from cytokines. In that sense, the sentence:

“The targeted genes were those encoding cytokines, chemokines, and interleukins, their receptors and signaling molecules, and genes involved in acute, chronic, and intracellular inflammatory responses (lines: 118-120)”,

should be modified like this:

“The targeted genes were those encoding cytokines (including chemokines and interleukins), their receptors and signaling molecules, and genes involved in acute, chronic, and intracellular inflammatory responses.

b) IL-18 receptor accessory protein (IL18RAP, or IL-18Rβ) is the other subunit or IL-18R, which means it is not a cytokine. In this context, it is correct to name together IL1B and IL18rap as pro-inflammatory molecules or factors than cytokines.

3. IL-18 is mistakenly written instead of IL-1B (line 199).

4. “Kyoto Encyclopedia of Genes and Genomics” (line 256) should be replaced by an abbreviation that has already been introduced.

5. Table 1. The prefix “up” is wrongly positioned, indicating Myd88 instead of Nfkb1.

Reviewer #2: In this manuscript by Uraguchi K et al titled “Upregulation of a nuclear factor-kappa B-interacting immune gene network mice cochleae with age-related hearing loss” , the authors have used gene expression analysis and bioinformatics analyses to demonstrate upregulation of inflammatory pathways in a C57BL/6 mouse model for ARHL. Further, they have used immunohistochemistry to show expression of IL-1B and IL18r1 in aged mouse cochlear tissue which is consistent with their gene expression analysis.

The experiments in this manuscript are technically sound and their conclusions are reasonable and these studies may be of interest for researchers trying to elucidate molecular basis for ARHL. A few minor comments:

1. It will be useful to have a list of up regulated and down regulated genes together with the volcano plot, which by itself is not very useful. The entire list of 84 genes on the array could be included in supplementary table.

2. Table 1 may need to be formatted- in the reviewer’s copy the “Regulation” column was not properly aligned with the gene name. For example for gene FASL, the “up” in the regulation column was not in the same row, this has happened for several genes.

3. Fig 4: the Y axis is not labeled

4. They need to carefully proofread the manuscript. For example:

Line 170: “The KEGG pathway is the annotation….” This sentence could be rephrased as “ The KEGG pathway is the annotation of functional gene pathways involving a group of genes”

Line 351: “The DEGs of these immune-related genes…” could be rephrased as “The differential expression of these immune-related genes…”

6. PLOS authors have the option to publish the peer review history of their article (what does this mean?). If published, this will include your full peer review and any attached files.

Reviewer #1: **Yes: **Sasa Vasilijic

Reviewer #2: No

---

## [Author Response · Author response to Decision Letter 0]

29 Aug 2021

August 28, 2021. PLOS ONE

Academic Editor

Dear Prof. Vasu Punj,

Thank you for inviting us to submit a revised manuscript entitled “ Upregulation of a nuclear factor-kappa B-interacting immune gene network in mice cochleae with age-related hearing loss“ to PLOS ONE.

We appreciate the time and energy dedicated by our reviewers toward improving our manuscript and are grateful for their thoughtful suggestions. Below, please find our responses to their comments.

In addition, in the previous decision Email of review process of this manuscript, we have been requested to present written permission from the copyright holder of Figure 5 as a Journal Requirement.

However, Figure 5 in the manuscript is our original figure from our own experiment and therefore we (or PLOS ONE journal) hold the copyright. This figure has not been published in any other publications.

Please also let us know if we have anything to do.

Comments from reviewer #1

Major comments

●Keeping that authors showed significant upregulation of NF-kB in the aged cochlear tissue and predicted its involvement in transcriptional regulation of eight significantly upregulated DEGs (Ccl5, Cxcl10, Cxcl9, Il1a, Lta, Ltb, Ptgs2, and Tnf), it is not clear why they did not show the expression of these genes on different ages (3-month-old, 6-month-old, 9-month-old and 12-month-old mice) as it was shown for nine genes selected based on the preliminary data from next-generation sequencing (RNAseq).

Response: We agree that it would be logically more consistent in our manuscript if we show the data of expression levels of the NF-kB-interacting genes on different ages (3-, 6-, 9-, and 12-month-old mice). In the process of progress in this study, we analyzed the expressions of the 9 genes (Casp1, IL18r1, IL18rap, IL1B, Card9, Clec4e, Ifit1, Ifit3, and Tlr9) with important immune functions, which are explained in the manuscript, before the time when we identified Ccl5, Cxcl10, Cxcl9, Il1a, Lta, Ltb, Ptgs2, and Tnf as the NF-kB-interacting upregulated genes in aged cochleae by the bioinformatic analyses.

The below sentences have been added as the limitation of study in the discussion.

“A limitation of this study is that the expression levels of the NF-kB-interacting upregulated genes (Ccl5, Cxcl10, Cxcl9, Il1a, Lta, Ltb, Ptgs2, and Tnf) were not analyzed by real-time RT-PCR at different ages (3-, 6-, 9-, and 12-month-old mice). Such data would provide important insights into the involvement of the NF-kB-interacting immune gene network in the molecular process of age-related hearing loss.”

●Although the authors referred in the discussion part that the hearing status is getting to worsen as animals are older, they should assess the hearing status of animals at all analyzed ages (at least at 7-weak, 6- month and 12- month age) in parallel with the gene expression analysis.

Response: According to this comment, we have performed additional experiments to examine hearing status by click-ABR thresholds (c-ABR thresholds) in 6-month-old mice. In the new Fig 1 of the revised manuscript, c-ABR thresholds in 6- to 7-week old, 6-month-old, and 12-month-old mice are shown. 

The below sentences are included in the result section of the revised manuscript.

“As shown in Fig 1, the c-ABR threshold was significantly higher in 6-month-old mice (63.0±14.9 dB SPL, mean±SD, n=10) and 12-month-old mice (66.9±13.6 dB SPL, n=8) than in 6–7-week-old mice (40±2.9, n=7) (P<0.01), confirming that 6- and 12-month-old mice exhibited significant age-related hearing loss.”

● Immunohistochemical analysis of cochlear sections from mice at different ages should be also extended for some of NF-kB -regulated significantly upregulated DEGs, as it was done for IL-18r1 and IL-1B.

Response: According to this comment, we have performed additional immunohistochemical experiments to visualize two NF-kB-interacting inflammatory molecules, TNFα (tumor necrosis factor α) and PTGS2 (prostaglandin-endoperoxide synthase 2) in mice cochleae (new Fig.6). TNFα and PTGS2 were expressed ubiquitously in the cochlear structures of the 12-month-old mice. Unequivocal immunoreactivity to TNFα and PTGS2 was observed in the lateral wall (the spiral ligament and stria vascularis) of aged cochleae.

●Considering the role of macrophages in the cochlear inflammatory response and the immune capacity of cochlear supporting cells, co-staining inflammatory mediators and cell-specific markers will contribute to identifying the primary cellular source of up-regulated cytokines and immune-related factors associated with ARHL.

Response: According to this comment, we have performed additional experiments to visualize macrophage/microglia-specific marker IBA1-positive cells in mice cochleae (new Fig 7). IBA1-positive macrophages were observed in the stria vascularis and the inferior division of spiral limbus in cochleae of 12-months old mice. No IBA1-positive cells were found in these cochlear structures of 6-week-old mice. (Double-immunofluorescent staining could not be performed by this immunohistochemistry using rabbit polyclonal antibody raised against IBA1, because we used rabbit-polyclonal antibodies against the inflammatory mediators.)

Minor comments

●Although authors called on their preliminary data, they should clarify how many genes were analyzed by next-generation sequencing (RNAseq). This statement will also put weight on the paper, considering that a similar study analyzing more than 700 genes has already been published.

Response: The below sentences were added to the “gene specific real-time RT-PCR” subsection of materials and methods; “In our preliminary data by RNA-seq, 800 genes were either upregulated (452 genes) or downregulated (348 genes) more than twofold in the aged cochleae of 12-month-old mice, compared with the cochleae of 6–7-week-old mice, and their functions were analyzed by bioinformatic analyses.”

●There is a terminology issue that should be solved.

Interleukins and chemokines are cytokines as well, and they should not be listed separately from cytokines. In that sense, the sentence:

“The targeted genes were those encoding cytokines, chemokines, and interleukins, their receptors and signaling molecules, and genes involved in acute, chronic, and intracellular inflammatory responses (lines: 118-120)”,

should be modified like this:

“The targeted genes were those encoding cytokines (including chemokines and interleukins), their receptors and signaling molecules, and genes involved in acute, chronic, and intracellular inflammatory responses.

Response: Thank you for the suggestion. The sentences were modified according to this comment.

●IL-18 receptor accessory protein (IL18RAP, or IL-18Rβ) is the other subunit or IL-18R, which means it is not a cytokine. In this context, it is correct to name together IL1B and IL18rap as pro-inflammatory molecules or factors than cytokines.

Response: IL1B and IL18rap are described as “pro-inflammatory molecules” throughout the text of the revised manuscript.

●IL-18 is mistakenly written instead of IL-1B (line 199).

Response: Thank you for the advice. The text was corrected according to this comment.

●“Kyoto Encyclopedia of Genes and Genomics” (line 256) should be replaced by an abbreviation that has already been introduced.

Response: The text was corrected according to this comment.

● Table 1. The prefix “up” is wrongly positioned, indicating Myd88 instead of Nfkb1.

Response: The positions of the texts in table 1 were reformatted to align appropriately with the gene name in each row in the revised table.

Comments from reviewer #2

Minor comments

●It will be useful to have a list of up regulated and down regulated genes together with the volcano plot, which by itself is not very useful. The entire list of 84 genes on the array could be included in supplementary table.

Response: Thank you for the suggestion. Table 1 has been modified to show a list of 33 upregulated and 7 downregulated genes. The entire list of 84 genes on the array is included in the supplementary table of the revised manuscript.

●Table 1 may need to be formatted- in the reviewer’s copy the “Regulation” column was not properly aligned with the gene name. For example for gene FASL, the “up” in the regulation column was not in the same row, this has happened for several genes.

Response: The positions of the texts in table 1 were reformatted to align appropriately with the gene name in each row in the revised table.

● Fig 4: the Y axis is not labeled.

Response: The Y-axis in Fig4 represents “fold change (gene expression levels relative to the levels in the young control cochleae). It is indicated in the new Fig 4 in the revised manuscript. 

●4. They need to carefully proofread the manuscript. For example:

Line 170: “The KEGG pathway is the annotation….” This sentence could be rephrased as “ The KEGG pathway is the annotation of functional gene pathways involving a group of genes”

Line 351: “The DEGs of these immune-related genes…” could be rephrased as “The differential expression of these immune-related genes…”

Response: The sentences were corrected in the revised manuscript according to this comment. The manuscript has been reviewed by a professional proofreading and editing service by a native English user.

Again, we appreciate the time and energy dedicated by our reviewers and editors. Changes in the revised manuscript are expressed in red characters.

We hope you would find the present manuscript would be suited for publication in PLOS ONE.

Sincerely yours,

Corresponding author

---

## [Decision Letter · Decision Letter 1]

11 Oct 2021

Upregulation of a nuclear factor-kappa B-interacting immune gene network in mice cochleae with age-related hearing loss

PONE-D-21-13350R1

Dear Dr. Maeda,

We’re pleased to inform you that your manuscript has been judged scientifically suitable for publication and will be formally accepted for publication once it meets all outstanding technical requirements.

Kind regards,

Vasu Punj

Academic Editor

PLOS ONE

Additional Editor Comments (optional):

Reviewers' comments:

Reviewer's Responses to Questions

**Comments to the Author**

1. If the authors have adequately addressed your comments raised in a previous round of review and you feel that this manuscript is now acceptable for publication, you may indicate that here to bypass the “Comments to the Author” section, enter your conflict of interest statement in the “Confidential to Editor” section, and submit your "Accept" recommendation.

Reviewer #1: All comments have been addressed

2. Is the manuscript technically sound, and do the data support the conclusions?

Reviewer #1: Yes

3. Has the statistical analysis been performed appropriately and rigorously? 

Reviewer #1: Yes

4. Have the authors made all data underlying the findings in their manuscript fully available?

Reviewer #1: Yes

5. Is the manuscript presented in an intelligible fashion and written in standard English?

Reviewer #1: Yes

6. Review Comments to the Author

Reviewer #1: (No Response)

7. PLOS authors have the option to publish the peer review history of their article (what does this mean?). If published, this will include your full peer review and any attached files.

Reviewer #1: **Yes: **Sasa Vasilijic

---

## [Editor Report · Acceptance letter]

13 Oct 2021

PONE-D-21-13350R1 

Upregulation of a nuclear factor-kappa B-interacting immune gene network in mice cochleae with age-related hearing loss 

Dear Dr. Maeda:

I'm pleased to inform you that your manuscript has been deemed suitable for publication in PLOS ONE. Congratulations! Your manuscript is now with our production department. 

Kind regards, 

on behalf of

Dr. Vasu Punj 

Academic Editor

PLOS ONE